# Isometric Shoulder Testing Using a Forcemeter Is a Reliable Method of Strength Evaluation

**DOI:** 10.3390/s23229106

**Published:** 2023-11-10

**Authors:** Joanna Wałecka, Przemysław Lubiatowski, Ewa Bręborowicz, Piotr Kaczmarek, Monika Grygorowicz, Leszek Romanowski

**Affiliations:** 1Rehasport Clinic, 60-201 Poznań, Poland; przemyslaw.lubiatowski@rehasport.pl (P.L.); piotr.kaczmarek@rehasport.pl (P.K.); monika.grygorowicz@rehasport.pl (M.G.); 2Department of Traumatology, Orthopaedics and Hand Surgery, University of Medical Sciences, 61-545 Poznań, Poland; ewabreborowicz@gmail.com (E.B.); romanowski.kcr@gmail.com (L.R.)

**Keywords:** isometric strength, shoulder strength, reliability, Forcemeter

## Abstract

Isometric strength testing using a digital dynamometer is reliable for muscle function evaluation. It allows us to objectify manual strength assessment measurement. We tested intra- and inter-observer reliability of a user-friendly efficient digital dynamometer—the Forcemeter—equipped with a computer program to monitor the measurements and to store the data. Abduction, forward flexion, and external and internal rotation of both shoulders were tested three times in 20 healthy volunteers with no record of shoulder trauma. Isometric contracture was recorded in newtons. The first and the third test were carried out by Examiner A (intra-rater reliability); the second test, by Examiner B (inter-rater reliability). Good reliability was shown for intra-class correlation coefficient (ICC) values which mean moderate to high correlations (r = 0.66–0.93) for both examiners. Moderate to high correlations (r = 0.72–0.91) were found for comparisons between the results obtained by Examiner A.

## 1. Introduction

Shoulder strength testing has been an important part of the patient’s functional evaluation. Apart from the subjective symptoms (pain, discomfort), range of motion (passive and active), and imaging, strength evaluation attempts to objectively reflect the patients’ ability to perform numerous activities necessary for daily living. This type of shoulder functioning can be affected by both the patients’ abilities (gender, age, body composition and profession, shoulder pain) and by the method of testing. There is a wide range of methods and approaches to measure shoulder strength. The simplest method is manual muscle testing, which does not require any instruments. Although this examination is easy to perform, it has been shown not to be replicable [1,2,3]. The final result is attributed to one of 6 levels of strength, which is subjectively judged by the examiner. Such examination can be made more objective by adding a simple dynamometer for an isometric muscle testing. This method was documented for the first time already in 1949 by Mayhew and Rothstein [4].

More advanced methods utilize sophisticated isokinetic or isotonic systems, which not only allow for a more dynamic evaluation (measurement of shoulder strength while it is in motion) but also for advanced calculations (the ability to perform work, assessing shoulder power and fatigue). The reliability of these dynamometers has been well documented (e.g., [5,6,7,8]). However, these devices have disadvantages: they are nonportable and require advanced computer software. A major drawback is also the high price of the device. In addition, the evaluation itself is more time-consuming and needs to be performed by well-trained personnel. It is, therefore, recommended mainly for biomechanical laboratories and sports trauma and rehabilitation centers. 

In contrast to fixed isokinetic dynamometers, isometric devices seem to be a balanced and objective compromise for testing shoulder strength. Users of these devices benefit from the fact that they are easy to use, portable, relatively inexpensive, and applicable in everyday practice (e.g., [9,10,11]). We have, therefore, found the necessity to use an isometric electronic dynamometer that would be controlled by personal computer software, enabling users to make multiple measurements, as well as record and archive all the results. The main justification of this study was the need to evaluate the clinimetric properties of such a device in order to present a new, portable isometric dynamometer for shoulder research as an alternative to isokinetic dynamometers, whose biggest disadvantages are their size, immobility, and the usage of significant electrical power.

Proper evaluation does not only result from the use of the device but from a proper, repetitive evaluation protocol. Therefore, the purpose of this study was to test the intra-rater and inter-rater reliability of the portable electronic isometric dynamometer Forcemeter in assessing shoulder forward flexion, abduction, as well as internal and external rotation in a group of healthy volunteers. 

## 2. Materials and Methods

### 2.1. Participants

A total of 20 healthy volunteers were enrolled in this study: 14 women and 6 men. The estimated sample size was based on guidance from Walter et al. who suggested that with 2 raters, a significance level of 0.05, and power of 80%, 19 samples were required to determine an ICC score of 0.7 [12]. The inclusion criteria were as follows: (i) age between 20–40 y.o.; (ii) normal shoulder function: full range of motion, no shoulder pain, negative tests, and (iii) level of activity: occasional or recreational sport. Individuals were excluded from this study if (i) a failure within the gleno–humeral joints was recorded during the examination, (ii) prior trauma, shoulder diseases, shoulder treatment, or neurological diseases were recorded, or (iii) active or professional sport experience appeared in the individual history. Any individuals with prior knowledge of the Forcemeter device were also excluded from this study in order to avoid possible bias and to justify the ease of use of the Forcemeter device for all future participants, regardless of their knowledge or lack of knowledge of the principle of operation of the device. All participants signed the informed consent. This study was approved by the Bioethical Committee of the University of Medical Sciences in Poznań.

### 2.2. Examiners

This study involved two examiners (A and B). Both examiners were experienced physiotherapists, well-trained in the field of shoulder assessment, and underwent an extensive training with the Forcemeter before this study.

### 2.3. Testing Device

The isometric evaluation of shoulder parameters was performed using an electronic isometric dynamometer (Forcemeter, Progress, Ostrów Wielkopolski, Poland). The Forcemeter consists of a PC control panel and a dynamometer. The PC control panel function is to interface the dynamometer with a personal computer. To communicate with the computer, the device is equipped with RS232 and a USB port. It is also the power supply for the system. The main element, which is the dynamometer, was attached to the base (the floor or the wall) by suction cups, and the muscle force was transferred to the device with the use of a non-elastic belt. The measuring range of the Forcemeter is 500 N (approx. ± 50 kG) with the accuracy of ±0.1% and the reading unit: 0.1 N. All of the results (which include participants’ and examiners’ data as well as the dynamometer measurements) were automatically recorded and stored using a dedicated PC Forcemeter software.

### 2.4. Testing Protocol

Participants were enrolled in three separate testing sessions for the following shoulder strength parameters: flexion, abduction, internal rotation, and external rotation. The first two tests were performed by Examiners A and B separately, in order to test the inter-rater reliability. After a seven day interval, participants took part in a third testing session, which was carried out by Examiner A and intra-rater reliability of the two tests was established.

The participants were instructed how to operate the device and performed several familiarization trials in a submaximal concentric mode before the data were collected. During examination, three trials were completed for each tested position with a minimum 20 s pause between trials. During the examination, the subjects were asked to produce a 6 s maximal contraction [13,14]. The greatest force produced was used as a measure of maximum muscular strength [N]. The mean values were calculated by the software as arithmetical average based on 5 sample measurements per second during muscle contraction. To minimize effects of muscle fatigue, at least 60 s of rest (as measured by the computer) was given between 20 sets. A longer, standard rest period was assigned before a second separate testing session performed by Examiner B, that is, one hour, after which the participant did not feel physically tired due to the examination. 

A standardized position was adopted for every examination. The individuals were always examined in a standing position. The dynamometer was fixed by suction cups to the floor (when shoulder flexion or abduction were tested) or to the wall (when internal or external shoulder rotation were tested). Flexion strength was evaluated with the arm in a 90° frontal deviation and the elbow in full extension. The arm was internally rotated (empty can—thumb down, Figure 1A). Shoulder abduction was tested with arm at a 90° deviation in the scapular plane (Figure 1B). In both cases, a non-elastic belt was looped around the distal forearm, 2 cm proximal to the radial styloid, and adjusted to the distance between the device and the forearm level (depending on participant’s height). Both internal (Figure 1C) and external rotation (Figure 1D) testing was performed with the arm along the side and elbow flexed to 90°. The non-elastic belt connected to the dynamometer was placed on the participant’s distal forearm, 2 cm proximal from radial styloid at adjusted level.

In order to test the reliability of the measurements, the results from both shoulders were analyzed together, regardless of the shoulder preference.

### 2.5. Statistical Analysis

Relative reliability was established for each shoulder position measurement (flexion, abduction, internal and external rotation), following the Shrout and Fleiss two-way random effect model (an absolute agreement definition) [15]. Analysis was performed among the 2 trials for one tester (intra-rater reliability) and between the 2 testers (inter-rater reliability). Single measure inter-class correlation coefficients (ICC 2.1 type) were calculated. We defined ICCs of less than 0.50 as indicative of poor reliability, values between 0.50 and 0.74 as indicative of moderate reliability, values between 0.75 and 0.90 as indicative of good reliability, and values greater than 0.90 as indicative of excellent reliability [16]. The precision of the individual measurements (the absolute reliability) was assessed with the Standard Error of Measurement (SEM). The sensitivity to change (the minimal amount of a change that a measurement must show to be greater than the within subject variability and measurement error) was calculated as Minimal Detectable Change (MDC) with 95% confidence (MDC95) [17]. To test the hypothesis that the values of a shoulder strength measured during two separate testing sessions performed by two different examiners or by one examiner with a 7-day interval are reliable, we calculated the F-distribution under the null hypothesis that there is a lack of an absolute concordance between the two raters performing the examination on the same day or between two measurements taken by one rater with a 7-day interval, respectively. All statistical calculations were performed using PQStat software for Windows, version 13.45.18 (PQStat Software, Poznań, Poland). The level of significance was set at *p* < 0.05.

## 3. Results

### 3.1. Intra-Rater Reliability of the Forcemeter

Results were correlated for both maximal and mean strength. Intra-rater reliability for the Forcemeter is presented in Table 1. The values of the inter-class correlation coefficients (ICCs) ranged from 0.65 to 0.95, with the lowest value for an internal rotation. Flexion testing proved to have excellent repeatability. This study has proven that measurement performed by the same examiner over a 7-day interval was consistent. Inter-class correlation coefficients were moderate to high for the parameters of maximum and mean force when measuring the compatibility for all evaluated movements. SEM was below 5.0 N and MDC was below 13.9 N for both tests by Examiner A (Table 1).

### 3.2. Inter-Rater Reliability of the Forcemeter

Inter-rater reliability for the Forcemeter device is presented in Table 2. The values of ICCs ranged from 0.70 to 0.91, with the lowest value for external rotation. Again, flexion testing achieved excellent reliability. By correlating the results obtained by two independent examiners, we found the force values to be also consistent. Inter-class correlation coefficients were moderate to high for the parameter of maximum force when measuring the compatibility between Examiners, for all evaluated movements. SEM was below 4.5 N and MDC was below 11.1 N for tests performed by Examiner A and B (Table 2).

## 4. Discussion

The most important finding of the present study is that measuring of the shoulder isometric strength with the Forcemeter proved that the device is highly reliable with respect to the determination of maximal and mean isometric shoulder strength.

Taking into consideration Kuhlman’s suggestions, all medical and clinical testing procedures should be comfortable and safe for patients in order to provide accurate and reproducible results [18]. Our study evaluated both the Forcemeter device as well as an isometric shoulder strength testing method. We have designed a specialized testing procedure for a reliable measuring of shoulder strength. The measurements were taken on the same day by two independent examiners and after a seven-day interval from the first testing session (both sessions by one examiner). The rationale for choosing the positioning of the testees was based on clinical relevance. In our study, we deliberately chose the position of the arm which resembles the position during regular clinical shoulder examination (e.g., Jobe Test, O’Brien’s Test). During our examinations, the testees were standing in natural positions, which further improved their comfort. We reasoned that that this issue is of special importance when testing problematic, dysfunctional, or painful shoulders. As was already previously reported, pain may cause some restrictions to the strength evaluation and results in technical difficulties, as it may be impossible to achieve 90 degrees of active range of shoulder abduction or flexion [7,19,20]. Moreover, a neutral position of the shoulder has been shown to be highly reliable for rotation measurements [18,21,22]. 

During the design of the testing protocol, we took special care to limit a possible interference from both the participants as well as from the examiners. It was already suggested that factors like a patient’s motivation, cooperation, learning effect, and fatigue do influence the muscle strength measurement results and the concentration of the rater, their experience, and accuracy in following the test procedures do influence the rater’s performance [23,24]. Measurements performed by less experienced [25] or physically weaker [23] raters do not reach satisfactory levels of reliability. Therefore, in our study, both examiners selected for the testing sessions were active, well-experienced physiotherapists, properly trained to use, and familiar with, the Forcemeter. Additionally, due to the fact that the Forcemeter can be attached either to the floor or to the wall, additional interference from the rater was excluded. 

It was recently suggested by Schrama et al. that the high risk of biased interpretation of test scores and their underestimated reliability can be attributed to unstable characteristics of the tests themselves [26]. We have, therefore, aimed at designing an unbiased isometric shoulder testing protocol, using a digital device, able to automatically gather, record, and store a prominent amount of data. We observed that the measurements taken on the same day by two independent examiners appeared to be highly reliable. Also, the measurements taken after a seven-day interval from the first testing session (both by one examiner) appeared to be highly reliable. Small values of SEM (<5.0 N) and MDC (<13.9) in our study indicate good absolute reliability. Comparable results have been reported in other examinations of reliability of isometric shoulder strength instruments [5,18,21,23,25,27,28]. In our study, the ICCs for flexion in both groups had excellent reliability. What is important to observe is that abduction and rotation had moderate to good ICCs, which may be caused by less stability of the trunk and stabilization of the rotation of the pelvis and trunk. 

What needs to be underlined is the underrated value of the Forcemeter in relation to already existing devices. To start, the average time to complete the whole examination in our study was 15 min. Comparing to the gold standard, the Biodex isokinetic device, it is significantly shorter. Biodex is more time-consuming: the preparation takes approximately 15 min before the actual measurements (additional 30–40 min) with Biodex can start. Moreover, in the present study, no negative effects were noted by the testees. None of the raters or testees complained about discomfort. Therefore, we concluded that both the comfort and the length of the examination are the two advantages of the Forcemeter. 

Except in cases of comfort and safety, the isometric testing in patients with affected shoulders is more reliable when compared to more advanced isokinetic testing (e.g., using Biodex), in which disabled shoulders cannot catch up with the set velocity of the testing device [29,30]. The Forcemeter is easy to apply and user-friendly because of its portability. Large devices are not mobile and difficult to carry. The Forcemeter is handy and can be used by physiotherapists and doctors in various offices, patient rooms or physiotherapy rooms, making the device easily available for any clinic or physiotherapy office. Also, the cost difference between the Forcemeter and isokinetic testing devices is prominent. A Biodex isokinetic device’s cost oscillates around USD 40,000 [31], whereas a Forcemeter costs approximately USD 1000 (manufacturer’s information). One should also consider ongoing costs: yearly maintenance and calibration is definitely more expensive for a Biodex than for Forcemeter. Therefore, in that aspect, our results indicate that a Forcemeter could be a preferential choice for shoulder evaluation as it is more affordable than isokinetic dynamometers.

However, we are aware of the existing limitations of our approach. Since the present study measured young, adult, healthy subjects, the results may differ from those of elderly subjects and those suffering from a disease. It should, therefore, always be taken into account that gender, age, and weight of testees might affect the results of shoulder strength test measurements [19,25].

Taking into consideration all that is mentioned above, we have concluded that isometric testing with a Forcemeter and an unbiased protocol offers a practical, reliable, and cost-effective alternative to more costly and time-consuming isokinetic testing when evaluating shoulder strength. A Forcemeter has not been evaluated in the literature before and its reliability is comparable to that of already well-known dynamometers. However, clear advantages of the device itself as well as of the proposed testing protocol make our approach highly recommended for assessing muscle strength in everyday medical practice and clinic.

## Figures and Tables

**Figure 1 sensors-23-09106-f001:**
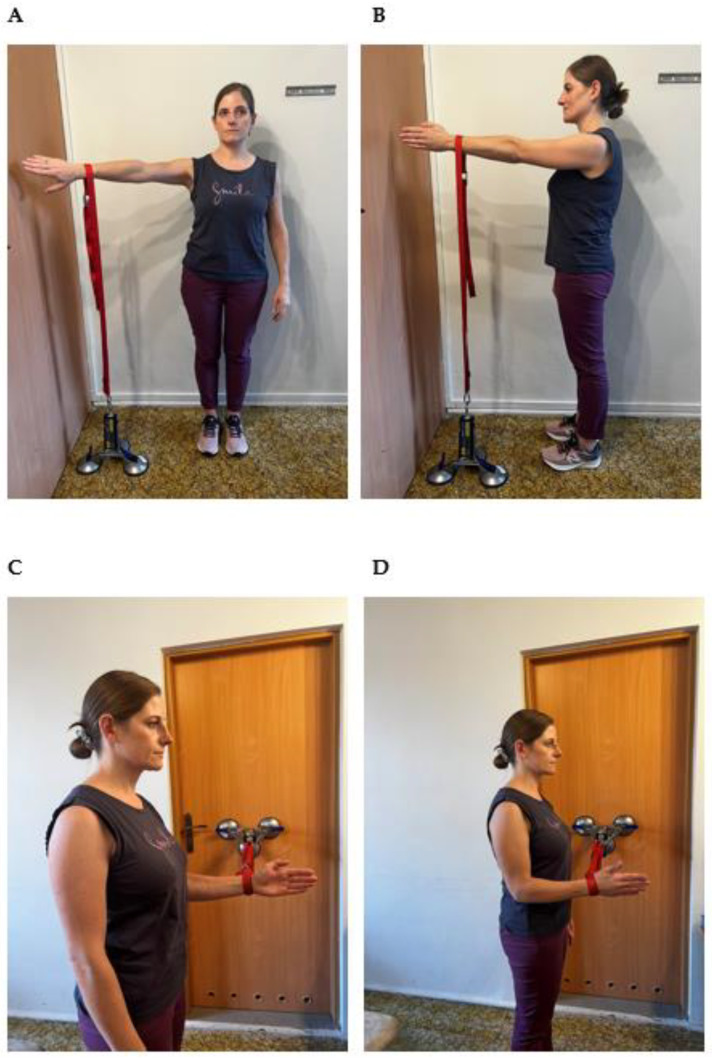
Isometric testing for abduction (**A**), flexion (**B**), internal rotation, (**C**) and external rotation (**D**).

**Table 1 sensors-23-09106-t001:** Intra-rater reliability of the isometric shoulder muscle strength measurements performed by Examiner A using a Forcemeter. T1—Test 1; T2—Test 2 (re-test); max—maximum value; mean—mean value; SEM—standard error of measurements; MDC—minimal detectable change with 95% confidence; ICC- Intra-Class Correlation coefficient.

	Flexion				Abduction				Internal Rotation				External Rotation			
	Max		Mean		Max		Mean		Max		Mean		Max		Mean	
	T1	T2	T1	T2	T1	T2	T1	T2	T1	T2	T1	T2	T1	T2	T1	T2
Mean [N]	70.0	72.3	58.5	59.6	60.0	64.4	58.2	53.4	95.0	98.2	79.3	81.8	79.9	80.7	68.1	67.3
SEM [N]	3.5	3.7	3.1	3.2	3.0	5.0	3.4	2.3	4.5	4.4	3.6	3.9	3.9	3.5	3.2	2.8
MDC [N]	9.7	10.3	8.6	8.9	8.3	13.9	9.4	6.4	12.5	12.2	10.0	10.8	10.8	9.7	8.9	7.8
ICC	0.95		0.91		0.71		0.70		0.65		0.70		0.81		0.74	

**Table 2 sensors-23-09106-t002:** Inter-rater reliability of the isometric shoulder muscle strength measurements performed by Examiners A and B using a Forcemeter. A—Examiner A; B—Examiner B; max—maximum value; mean—mean value; SEM—standard error of measurements; MDC—minimal detectable change with 95% confidence; ICC—Intra-Class Correlation coefficient. The mean values of the maximum and mean values from two tests (T1, T2) for Examiner A are presented.

	Flexion				Abduction				Internal Rotation				External Rotation			
	Max		Mean		Max		Mean		Max		Mean		Max		Mean	
	A	B	A	B	A	B	A	B	A	B	A	B	A	B	A	B
Mean [N]	71.1	68.9	59.0	60.2	62.2	63.9	55.8	54.3	96.6	89.1	80.5	74.9	80.3	82.4	67.7	70.8
SEM [N]	3.5	3.8	3.1	3.3	3.0	3.5	3.4	3.0	4.5	4.0	3.6	3.3	3.9	4.3	3.2	3.4
MDC [N]	9.7	10.5	8.6	9.1	8.3	9.7	9.4	8.3	12.5	11.1	10.0	9.1	10.8	11.9	8.9	9.4
ICC	0.91		0.87		0.78		0.77		0.75		0.74		0.74		0.70	

## Data Availability

Data are contained within the article.

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
