# Peer review of "Isometric Shoulder Testing Using a Forcemeter Is a Reliable Method of Strength Evaluation"

_sensors, 2023, doi:10.3390/s23229106_

Round 1
Reviewer 1 Report
Comments and Suggestions for Authors
Dear authors,
I have not identified the main justification nor the relevance of the present study. In addition, reliability is a measurement property different from validity. I would recommend following the COSMIN (COnsensus-based Standards for the selection of health Measurement INstruments) aimed at improving the present research.
Comments on the Quality of English LanguageExtensive editing of English and language required.
Author Response
Dear authors,
I have not identified the main justification nor the relevance of the present study.
Our response: We have introduced the explanatory sentences in the Introduction section, as suggested.
In addition, reliability is a measurement property different from validity. I would recommend following the COSMIN (COnsensus-based Standards for the selection of health Measurement INstruments) aimed at improving the present research.
Our response: Thank you for that comment. We indeed followed the guidelines of the COSMIN, which defines the reliability as “the proportion of the total variance in the measurements which is due to true differences between patients” (Prinsen CAC, Mokkink LB, Bouter LM, Alonso J, Patrick DL, de Vet HCW, Terwee CB. COSMIN guideline for systematic reviews of patient-reported outcome measures. Qual Life Res. 2018 May;27(5):1147-1157.) We have therefore tested test the intra-rater and inter-rater reliability of the device with the means of intraclass correlation coefficient (ICC), following the general guidelines, as reported in Koo TK, Li MY. A Guideline of Selecting and Reporting Intraclass Correlation Coefficients for Reliability Research. J Chiropr Med. 2016 Jun;15(2):155-63. Following the Rewiever’s suggestion, we carefully went through the text to avoid incorrect use of the words “validity” or “validate” and changed them accordingly.
Extensive editing of English and language required.
Our response: The manuscript underwent English revisions.
Reviewer 2 Report
Comments and Suggestions for Authors
Thank you for this opportunity to review this paper on the repeatability of the Forcemeter. A standardized, portable strength measuring device can offer better insights to many different populations (poor, weak, immobile, quick tests in clinics, etc.). I hope my questions and comments will help strengthen this manuscript.
Abstract:
1. Examiner A should be intra-rater reliability.
2. To be consistent with existing literature, please replace “medium” with “moderate” when discussing the level of correlations.
Introduction:
3. The authors might consider leveraging the Forcemeter as an alternative to one of the biggest disadvantages of the large, immobile dynamometers also can require significant electrical power. Though I could not find it on a websearch, from the description and picture, it looks as though the Forcemeter does not require much electricity.
Methods:
4. Why were people with knowledge of the Forcemeter excluded?
5. What was the type of extensive training required by the examiners to operate the Forcemeter? Were the directions/script each examiner gave standardized?
6. Were there any ceiling effects for the 500N range of the Forcemeter? Did participants all measure below the max?
7. How much rest was given between examiners? Was it standard for all participants or did it vary?
8. Was the order of the examiners randomized for each participant? If so, please state in the methods. If not, please explain why the authors feel the results were not affected by this or add it to the limitation section.
Results:
9. SEM and MDC for T1 in Table 1 match Examiner A’s values in Table 2, but why are the mean/max values not matching for T1 and A in the two tables?
Discussion:
10. (Pg 6, line 206) Was the Forcemeter attached to the wall? Figure 2 shows ground for abduction and flexion. How were the rotations measured? Can this set-up be added to Figure 2?
11. (Pg 6, lines 234-237) This statement needs a reference that isometric is more reliable than isokinetic.
12. (Pg 6, lines 240+) This paper reported reliability in healthy volunteers. That should not be extrapolated into a diseased population. Repeatability of the diseased population is necessary as pain changes biomechanics, reliability, and repeatability.
13. The limitation paragraph should not end the paper. The take-away message is lost in the jumble of the discussion. The Forcemeter is repeatable is understood. Pg 6, line 225-227 makes a better conclusion. The two paragraphs after can be incorporated better throughout the discussion.
Comments on the Quality of English LanguageThe authors use intraclass intra class and interclass correlations coefficients interchangeably (for example on pages 4-5).
Keep results and discussion of results in past tense.
The authors should change "his/her" to "their" as their is now recognized as a singular pronoun (Pg 5 line 201).
Page 6, lines 218-220 are very hard to read and understand. I would suggest: "In our study, ICC for flexion in both groups had excellent reliability. What is important to observe is that adduction and rotation had moderate to good ICC which may be caused by less stability ..."
Page 6, line 229: "mentioned" should replace "motioned" and "it is" should replace "its".
Author Response
Thank you for this opportunity to review this paper on the repeatability of the Forcemeter. A standardized, portable strength measuring device can offer better insights to many different populations (poor, weak, immobile, quick tests in clinics, etc.). I hope my questions and comments will help strengthen this manuscript.
Abstract:
- Examiner A should be intra-rater reliability.
Our response: We have changed the word as suggested .
- To be consistent with existing literature, please replace “medium” with “moderate” when discussing the level of correlations.
Our response: We do agree with that and changed as suggested .
Introduction:
- The authors might consider leveraging the Forcemeter as an alternative to one of the biggest disadvantages of the large, immobile dynamometers also can require significant electrical power. Though I could not find it on a websearch, from the description and picture, it looks as though the Forcemeter does not require much electricity.
Our response: Following the Reviewer’s suggestion, we have added some comments considering the Forcemeter obvious advantages both in the Introduction and the Discussion sections.
Methods:
- Why were people with knowledge of the Forcemeter excluded?
Our response: One of the aims of the study was to examine the ease of use of the Forcemeter device for all future participants, regardless of their knowledge or lack of knowledge of the principle of operation of the device. We have reasoned that if the respondents were familiar with the device, they might question the simplicity of its use. Thus we decided not to include participants with previous knowledge on the Forcemeter to avoid the possible bias. In our opinion such situation would reflect its clinical use. We have added an appropriate short explanation in the Methods section.
- What was the type of extensive training required by the examiners to operate the Forcemeter? Were the directions/script each examiner gave standardized?
Our response: As we stated in the Methods section, both examiners underwent an extensive training with the Forcemeter device before the study. Our assumption was that the examiner has to evaluate the participants easily, quickly and repeatedly. The need for examiners’ training on how to operate the device before the study was justified by the need to perform tests on participants according to the same, standardized protocol.
- Were there any ceiling effects for the 500N range of the Forcemeter? Did participants all measure below the max?
Our response: we did not observe any ceiling effects.
- How much rest was given between examiners? Was it standard for all participants or did it vary?
Our response: It was a standard time of one hour, after which the participant did not feel physically tired due to the examination. We have added this explanatory sentence to the Methods section.
- Was the order of the examiners randomized for each participant? If so, please state in the methods. If not, please explain why the authors feel the results were not affected by this or add it to the limitation section.
Our response: The order of the examiners was not randomized.
Results:
- SEM and MDC for T1 in Table 1 match Examiner A’s values in Table 2, but why are the mean/max values not matching for T1 and A in the two tables?
Our response: The mean and max values for Examiner A presented in Table 2 are the mean values from two measurements (both values are presented in Table 1). We have added an explanatory sentence in the Table 2 legend.
Discussion:
- (Pg 6, line 206) Was the Forcemeter attached to the wall? Figure 2 shows ground for abduction and flexion. How were the rotations measured? Can this set-up be added to Figure 2?
Our response: In the present study the Forcemeter was attached to the floor, as presented in Figure 2. However, there is a possibility of attaching the device to the wall, which we mentioned in the Discussion section, however we did not perform the measurements with the Forcemeter fixed to the wall.
- (Pg 6, lines 234-237) This statement needs a reference that isometric is more reliable than isokinetic.
Our response: Following the Reviewer’s suggestion, we have including references to two PhD theses.
- (Pg 6, lines 240+) This paper reported reliability in healthy volunteers. That should not be extrapolated into a diseased population. Repeatability of the diseased population is necessary as pain changes biomechanics, reliability, and repeatability.
Our response: Following the Reviewer’s suggestion, we did not discuss the data published in reference [19] here.
- The limitation paragraph should not end the paper. The take-away message is lost in the jumble of the discussion. The Forcemeter is repeatable is understood. Pg 6, line 225-227 makes a better conclusion. The two paragraphs after can be incorporated better throughout the discussion.
Our response: Following the Reviewer’s suggestion, we have changed the course of the discussion to conclude with the sentences which were originally in lines 225-227.
The authors use intraclass intra class and interclass correlations coefficients interchangeably (for example on pages 4-5).
Our response: Following the Rewiever’s suggestion, we carefully went through the text to avoid incorrect use of the words and changed them accordingly.
Keep results and discussion of results in past tense.
Our response: Following the Rewiever’s suggestion, we carefully went through the text to and changed the results and discussion of results in past tense.
The authors should change "his/her" to "their" as their is now recognized as a singular pronoun (Pg 5 line 201).
Our response: We have changed that, as suggested.
Page 6, lines 218-220 are very hard to read and understand. I would suggest: "In our study, ICC for flexion in both groups had excellent reliability. What is important to observe is that adduction and rotation had moderate to good ICC which may be caused by less stability ..."
Our response: We have changed that, as suggested.
Page 6, line 229: "mentioned" should replace "motioned" and "it is" should replace "its".
Our response: We have changed that, as suggested.
Reviewer 3 Report
Comments and Suggestions for Authors
See the attached file.

Author Response
The paper presents a very thorough experimental study of the use of a commercial device for the testing of the isometric shoulder strength carried out on a lot of healthy patients.
The setup of the study, the measurement protocol and the entire data interpretation are quite solid and seem to be quite reliable and accurate. However, there are several aspects which should be integrated into the paper to put emphasis in a better way on the added value of the listed device and a more detailed comparison with other existing solutions.
First of all, I would begin with the targeted patients and their disorders. You are targeting only patients with muscle trauma or also patients which have disorders of neurological causes?
Our response: In the present study, we have enrolled only healthy volunteers with normal shoulder function. This is the first study to evaluate the Forcemeter utility in isometric shoulder testing. We have underlined that in our manuscript. However, due to the positive outcome of the study, we definitely plan to use the device in future clinical trials. Despite such clinical evaluation is far beyond this paper, we might assume that the target population would include both, muscle trauma and neurological patients with impaired shoulder function or after a shoulder trauma, provided, however, that they are able to perform the given movements.
For neuromotor deficit, can you replicate the measurement in case the subject has difficulties in
maintaining a stating posture?
Our response: In the present study we have evaluated the repeatability of the Forcemeter in a standing position solely. Therefore so far the examination is limited to the participants which are able to perform the given movements in a standing position. However, as a future direction, taking into consideration our patients needs and comorbidities, we plan to test the repeatability of the Forcemeter in a sitting position.
Why do you state in the introduction that the shoulder functioning is dependent on the method of
testing? Are there any medically approved tests which can lead to an acute muscle disorder at the level of the shoulder?
Our response: We stated that the shoulder functioning is dependent on the method of testing to underline that the examination position may affect the shoulder performance during the test. Hence, it is important to perform quality assurance tests in the correct position, that is exactly the same position for some resisting isometric movement. This issue has been evaluated and reported by e.g.
Itoi E, Kido T, Sano A, Urayama M, Sato K. Which is more useful, the "full can test" or the "empty can test," in detecting the torn supraspinatus tendon? Am J Sports Med. 1999 Jan-Feb;27(1):65-8.
or
Lee CK, Itoi E, Kim SJ, Lee SC, Suh KT. Comparison of muscle activity in the empty-can and full-can testing positions using 18 F-FDG PET/CT. J Orthop Surg Res. 2014 Oct 1;9:85.
To answer the second part of the question, the medical test cannot be approved if it can lead to any disorders, including an acute muscle disorder. It is of general acceptance that the clinical test are safe to perform for participants.
What is the real end-target of these measurements?
Our response: The end-target of the measurements performed with the Forcemeter is the repeatability, as we stated in the manuscript.
There is another statement in the paper regarding the difficulties in using different apparatus in terms of time and costs, but they are not accurately detailed.
Thus, after defining the targeted population, I would suggest a critical analysis of different devices interms of costs, efficiency, measurement time, specific benefits, advantages, drawbacks. In terms of costs which do you consider an “expensive” device? Are there any limitations for the Forcemeter compared to other “more expensive” devices?
Our response: Following the Reviewer’s suggestion, we’ve added this missing information considering pros and cons of the Forcemeter in relation to other existing devices, based on the published data and our clinical experience with different devices.
There is no mention of the accuracy of the Forcemeter. The maximum force is given but the lowest
detectable value or the resolution of the device is not specified. In case of a patient with a shoulder trauma or a healthy young female (no bias here on gender issue, but statistically women have lower muscle strength or maximum generated force compared to males) which has a “natural” maximum force in the muscle shoulder of 150 N?
Having to operate in the first third of the scale is influencing the results?
How do you classify this? In case of unilateral disorder, you might consider the difference
between left and right sides, but what if you have a slight bilateral disorder?
Our response: The accuracy of the Forcemeter, according to the manufacturer is ±0,1% and the reading unit: 0,1N. We have added this missing information to the revised version of the manuscript.
The Forcemeter can measure the strength within the whole range of the scale (0-500N) with the same accuracy, therefore, operating in the first third of the scale has no impact on the results.
In all well-planned clinical comparisons, impaired shoulder function is compared to the shoulder function of healthy participants of the same age, high and BMI, therefore even bilateral disorders could be assessed and compared. Moreover, patients after surgical procedures or trauma, could be evaluated at multiple follow-ups and the results could be compared to the baseline levels.
While the material presented in the paper is very well written I consider that more information must be added to justify well-documented research. As no comparison is done with other devices one could even suspect (even though I don’t think this is the case) that this is an “advertise material” for a certain commercial product.
There is a lot of potential in this paper but it needs improvement.
Our response: We are grateful for all of the comments. We have introduced the changes specified above, all of them helped to improve our manuscript.
Round 2
Reviewer 2 Report
Comments and Suggestions for Authors
Thank you for taking the time to thoughtfully address my comments and the opportunity to re-review this paper. The changes have made this manuscript much clearer. Though most of my questions and comments were fully addressed, there were three that still were not fully answered.
Methods:
- Why were people with knowledge of the Forcemeter excluded?
Our response: One of the aims of the study was to examine the ease of use of the Forcemeter device for all future participants, regardless of their knowledge or lack of knowledge of the principle of operation of the device. We have reasoned that if the respondents were familiar with the device, they might question the simplicity of its use. Thus we decided not to include participants with previous knowledge on the Forcemeter to avoid the possible bias. In our opinion such situation would reflect its clinical use. We have added an appropriate short explanation in the Methods section.
Response to rebuttal #4: Participants knew about the Forcemeter the second test and the second day of testing. How does this affect your repeatability?
Results:
- SEM and MDC for T1 in Table 1 match Examiner A’s values in Table 2, but why are the mean/max values not matching for T1 and A in the two tables?
Our response: The mean and max values for Examiner A presented in Table 2 are the mean values from two measurements (both values are presented in Table 1). We have added an explanatory sentence in the Table 2 legend.
Response to rebuttal #9: Have the authors consulted a statistician regarding this approach? How do the results differ if T1 A was compared to B? Why would the SEM and MDC of A match T1 and not T1 + T2?
Discussion:
- Pg 6, line 206) Was the Forcemeter attached to the wall? Figure 2 shows ground for abduction and flexion. How were the rotations measured? Can this set-up be added to Figure 2?
Our response: In the present study the Forcemeter was attached to the floor, as presented in Figure 2. However, there is a possibility of attaching the device to the wall, which we mentioned in the Discussion section, however we did not perform the measurements with the Forcemeter fixed to the wall.
Response to rebuttal #10: Pg 4 lines 119-120 – This clearly states that the Forcemeter was attached to the wall for internal and external rotations. Internal and external rotations are presented in the results. It remains unclear how the rotations were measured without a diagram/picture.On page 4, lines 128-129, can the authors state where on the floor relative to the participant with their arm bent at 90 degrees the Forcemeter was placed or add a picture of internal/external rotation measurements?
Author Response
Thank you for taking the time to thoughtfully address my comments and the opportunity to re-review this paper. The changes have made this manuscript much clearer. Though most of my questions and comments were fully addressed, there were three that still were not fully answered.
Methods:
- Why were people with knowledge of the Forcemeter excluded?
Our response: One of the aims of the study was to examine the ease of use of the Forcemeter device for all future participants, regardless of their knowledge or lack of knowledge of the principle of operation of the device. We have reasoned that if the respondents were familiar with the device, they might question the simplicity of its use. Thus we decided not to include participants with previous knowledge on the Forcemeter to avoid the possible bias. In our opinion such situation would reflect its clinical use. We have added an appropriate short explanation in the Methods section.
Response to rebuttal #4: Participants knew about the Forcemeter the second test and the second day of testing. How does this affect your repeatability?
Our response: We reasoned that just one prior examination with the Forcemeter is not the same as well-established, good knowledge of the Forcemeter action. We have therefore excluded potential participants with good knowledge of the Forcemeter solely, and assumed that one examination would not provide biases to the second examination.
Results:
- SEM and MDC for T1 in Table 1 match Examiner A’s values in Table 2, but why are the mean/max values not matching for T1 and A in the two tables?
Our response: The mean and max values for Examiner A presented in Table 2 are the mean values from two measurements (both values are presented in Table 1). We have added an explanatory sentence in the Table 2 legend.
Response to rebuttal #9: Have the authors consulted a statistician regarding this approach? How do the results differ if T1 A was compared to B? Why would the SEM and MDC of A match T1 and not T1 + T2?
Our response: Thank you for your close inspection of the data. Yes, indeed, we did consult these values with the statistician, who confirmed the correctness of presented data.
Discussion:
- Pg 6, line 206) Was the Forcemeter attached to the wall? Figure 2 shows ground for abduction and flexion. How were the rotations measured? Can this set-up be added to Figure 2?
Our response: In the present study the Forcemeter was attached to the floor, as presented in Figure 2. However, there is a possibility of attaching the device to the wall, which we mentioned in the Discussion section, however we did not perform the measurements with the Forcemeter fixed to the wall.
Response to rebuttal #10: Pg 4 lines 119-120 – This clearly states that the Forcemeter was attached to the wall for internal and external rotations. Internal and external rotations are presented in the results. It remains unclear how the rotations were measured without a diagram/picture.On page 4, lines 128-129, can the authors state where on the floor relative to the participant with their arm bent at 90 degrees the Forcemeter was placed or add a picture of internal/external rotation measurements?
Our response: Following the Reveiwer’s suggestion, we took additional pictures of the Forcemeter examination of internal and external rotation and changed the Figure 1 in the revised version of the manuscript.
Reviewer 3 Report
Comments and Suggestions for Authors
The authors have responded in a very good manner to the observations/recommendations and thus I consider now that the paper can be published. There is one minor observation, but this will be clearly solved during the final editing (there is a blank page in the paper - page 3)
Author Response
The authors have responded in a very good manner to the observations/recommendations and thus I consider now that the paper can be published. There is one minor observation, but this will be clearly solved during the final editing (there is a blank page in the paper - page 3)
Our response: Thank you for your effor reviewing our manuscript, we have now edited the manuscript to avoid the blank page.
Round 3
Reviewer 2 Report
Comments and Suggestions for Authors
Thank you for answering my extensive questioning. I have no further comments/questions.